# MARFT: Multi-Agent Reinforcement Fine-Tuning

## Abstract

The rapid rise of LLM-based agents has led to the emergence of LLM-based Multi-Agent Systems (LaMAS), which show strong potential in complex, collaborative tasks such as presentation generation and even scientific research. While Reinforcement Learning is well-established in enhancing LLM-based agent performance, its success has largely focused on single-agent settings. In contrast, applying Multi-Agent Reinforcement Learning to LaMAS remains limited. This is due to fundamental mismatches between traditional MARL assumptions and the unique dynamics of LaMAS, including action asynchronicity, dynamic organization, and characteristic profiles, which present significant new challenges. To address these challenges, we first formalize LaMAS optimization as a Flex-MG, capturing agent heterogeneity and interdependence, and then propose a novel paradigm termed **Multi-Agent Reinforcement Fine-Tuning (MARFT)**, introducing a new optimization framework for LaMAS. Two naive instantiations of MARFT are implemented on the action-level and token-level. Comparative experiments demonstrate MARFT's superior stability and performance over representative methods, while extensive ablation studies and analysis on math problem-solving and coding benchmarks further validate its effectiveness and efficiency, establishing it as a principled and generalizable approach for tuning LaMAS. As this work establishes a new paradigm, we conclude by highlighting the limitations of current research and pinpointing promising directions for future work.

## 1 Introduction

Large Language Models (LLMs) are increasingly being deployed as autonomous agents capable of decision-making, reasoning, and interacting with complex, dynamic environments (Jin et al., 2024; Hong et al., 2024; Qian et al., 2024). Beyond their advanced natural language capabilities (Chowdhary, 2020), LLMs support retrieval-augmented generation (RAG) (Lewis et al., 2021) and, when integrated with external tools or APIs, can perform sophisticated tasks across computing platforms (Erdogan et al., 2024; Zhang et al., 2025b). They have also been embedded in embodied and simulated environments as agents in robotics and gaming (Tan et al., 2024; Carta et al., 2023). Their capacity to follow instructions, learn from feedback, and generate context-aware responses supports applications in healthcare, education, and software development (Dai et al., 2023; Chen et al., 2024).

Multi-Agent Systems (MAS), particularly LLM-based MAS (LaMAS), have consistently outperformed single-agent models in complex tasks, as evidenced by the GAIA leaderboard (Mialon et al., 2024), where top-performing systems are multi-agent frameworks. Leading platforms such as OpenAI Agents SDK (formerly Swarm), Microsoft AutoGen (Wu et al., 2023), Magnetic-One, CAMEL-AI OWL (CAMEL-AI.org, 2025), and Google's AI Co-Scientist (Gottweis et al., 2025) highlight the growing significance of this architecture. OpenAI's AGI roadmap further positions MAS at the apex of its system hierarchy. However, researchers (Cemri et al., 2025) identify that MAS face fourteen distinct failure modes tied to design, coordination, and control—areas where MARL holds promise. Additionally, Yang et al. (2025b) introduced the first framework to systematically evaluate the contributions of individual modules, paving the way for more efficient MAS optimization.

Recent advances in tuning of multiple LLM-based agents can be broadly categorized into two approaches: *tuning-free* and *parameter fine-tuning*. Tuning-free methods avoid altering agent pa-

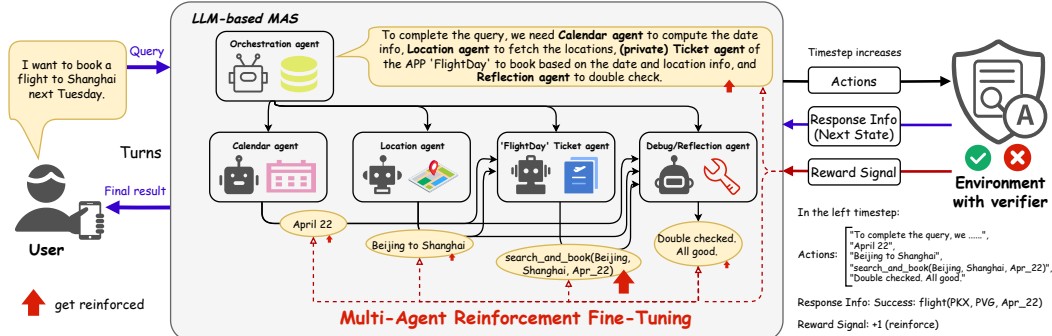

Figure 1: Illustration of MARFT in real-world agentic problem-solving scenarios.

rameters, instead relying on strategies like prompt engineering (Fernando et al., 2023), in-context learning (Tao et al., 2024), and self-evolution mechanisms (Yang et al., 2025a; Huang et al., 2024), which iteratively refine prompts or agent profiles. Parameter fine-tuning directly modifies model parameters or architectures, employing techniques such as multi-agent debate to generate training data (Subramaniam et al., 2025), or fine-tuning specialized modules for tasks like code synthesis (Khattab et al., 2024). An innovative example duplicates an LLM into pioneer and observer roles to enable mutual learning via knowledge transfer and role switching (Ma et al., 2024). However, these methods are constrained by their reliance on a supervised learning paradigm, which is fundamentally misaligned with the sequential decision-making nature of agent tasks. MARTI (Zhang et al., 2025a) claims to be an RL framework for LaMAS. However, it still uses separate policy trainers, which is like independent PPO without explicit collaboration and treating each agent as a separate entity. The most related work, MAPoRL (Park et al., 2025), rooted in MARL, pioneers in applying multi-agent PPO to post-train LLMs, but does not consider the unique characteristics of LaMAS and assumes homogeneous agent behavior through collaborative debate, diverging from real-world scenarios where agents often perform more complicated and diverse forms of collaboration, beyond simple debate, to solve complex problems.

To address these limitations and fill the research gaps, we propose **Multi-Agent Reinforcement Fine-Tuning (MARFT)**, a novel paradigm designed to advance the capability and coordination of LaMAS, and fully unleash the potential of LaMAS. Figure 1 gives a manifestation of MARFT in real-world agentic problem-solving scenarios. Unlike existing approaches, MARFT treats agents as an integrated, heterogeneous system that learns through interaction and reinforcement rather than static supervision. Our key contributions are:

- **Flex-MG:** A formalization of the novelly proposed Flex-MG, which uniquely incorporates a dynamic dependency function to formally model the fluid, conditional interdependencies between agents, thereby providing a crucial theoretical framework for fine-tuning LaMAS.

- **MARFT Paradigm:** A detailed introduction and interpretation of the new paradigm MARFT, and a universal algorithmic framework, seamlessly integrated with LaMAS, supported by the established MARL theories.

- **Concrete Algorithms:** Proposal of MARFT-A and MARFT-T as two concrete instantiations of MARFT and a series of experiments on mathematical and coding problems to validate the effectiveness of MARFT.

- **Foundational Research Avenues:** A detailed discussion of limitations and open questions, which pinpoints the future research directions that are essential for theoretical and practical advancement in reinforcement fine-tuning LaMAS beyond MARFT.

## 2 RELATED WORKS

### 2.1 LLMS AS AGENTS

Recent advances in large language models (LLMs) have enabled the development of autonomous agents with capabilities in complex reasoning, planning, and decision-making (Luo et al., 2025;

Yao et al., 2023; Shinn et al., 2023; Wang et al., 2023). These agents incorporate components such as memory, tool use, and planning to perform tasks across diverse domains (Schick et al., 2023; Tang et al., 2023; Erdogan et al., 2024; Lin et al., 2025). Frameworks like AutoGen (Wu et al., 2023) exemplify this integration by enabling multi-agent collaboration between LLMs, humans, and external tools. Recent surveys outline unified architectures featuring perception, memory, planning, and action modules, showcasing the versatility of LLM agents in fields from social sciences to engineering (Wang et al., 2024). Nonetheless, significant challenges remain in achieving long-term autonomy, adaptability to dynamic environments, and robust decision-making (Du et al., 2025).

## 2.2 LLM-BASED MULTI-AGENT SYSTEMS (LaMAS)

The concept of LaMAS has recently gained prominence (Guo et al., 2024; Yang et al., 2024b), with numerous applications and inference frameworks emerging to address complex agentic problems (Epperson et al., 2025). Unlike the multi-agent team in traditional MARL, where agents act synchronously with uniform decision weights, LaMAS introduces a highly dynamic organization and asynchronous execution. Agents in LaMAS can dynamically decompose tasks, adapt workflows based on execution dependencies, and coordinate actions in a decentralized yet goal-aligned manner.

While most existing optimization techniques for LaMAS are parameter-free (Yang et al., 2025a; Tao et al., 2024; Huang et al., 2024; Fernando et al., 2023), recent efforts aim to enhance its capabilities through efficient parameter-tuning. Multiagent Finetuning (Subramaniam et al., 2025) improves LLMs via collaborative debate among specialized agents, outperforming single-agent self-improvement in performance. CORY (Ma et al., 2024) uses two coevolving agents that alternate roles during fine-tuning. The most related work, MAPoRL (Park et al., 2025), while rooted in MARL, assumes homogeneous agent behavior through collaborative debate, diverging from real-world scenarios where agents often perform diverse or orthogonal roles to solve complex problems.

## 2.3 MULTI-AGENT REINFORCEMENT LEARNING (MARL)

Though there exist lots of value-based MARL methods, such as QMIX (Rashid et al., 2018), VDN (Sunehag et al., 2017), those based on Proximal Policy Optimization (PPO) (Schulman et al., 2017) are more applicable to LLMs, have been shown to be one of the most effective RL methods for decision-making tasks, and give a guarantee of monotonic improvement (Nakibinge et al., 2024; Ouyang et al., 2022; Heess et al., 2017; Schulman et al., 2015).

**IPPO, MAPPO, and HAPPO** Independent PPO (IPPO) and Multi-Agent PPO (MAPPO) Yu et al. (2022) are early approaches applying PPO in multi-agent settings. Both use standard PPO per agent; IPPO assigns each agent its own value function, while MAPPO uses a shared central critic. However, MAPPO assumes parameter sharing across agents and lacks guarantees of monotonic team reward improvement, as local policy updates can degrade overall performance. HATRPO and HAPPO address this by sequential policy updates (Kuba et al., 2022).

**MAT** Multi-Agent Transformer (Wen et al., 2022) is a novel MARL paradigm from the perspective of multi-agent sequential decision-making, re-modeling MARL problems into Sequential Modeling (SM) problems via Multi-Agent Advantage Decomposition Theorem. Within this paradigm, the approach to approximating the value function remains consistent with conventional MARL methods, but for the policy optimization, it utilizes a modified clipping PPO objective. MAT assumes that, although agents act simultaneously, they make decisions sequentially in an arbitrary order, based on the planned decisions of their predecessors.

## 3 EXTENDING MARL FOR LaMAS OPTIMIZATION

LaMAS differs significantly from traditional multi-agent systems in MARL, making large-scale implementation within standard MARL frameworks challenging. Though there is little to no research applying existing MARL methods, such as COMA (Foerster et al., 2018), MADDPG (Lowe et al., 2017), MAPPO (Yu et al., 2022), and HAPPO (Kuba et al., 2022), to LaMAS, the theoretical studies and effectiveness of these methods in conventional MARL problems illuminate our path to extend MARL for LaMAS. Here, we formalize a novel Flex-MG to help address the highly

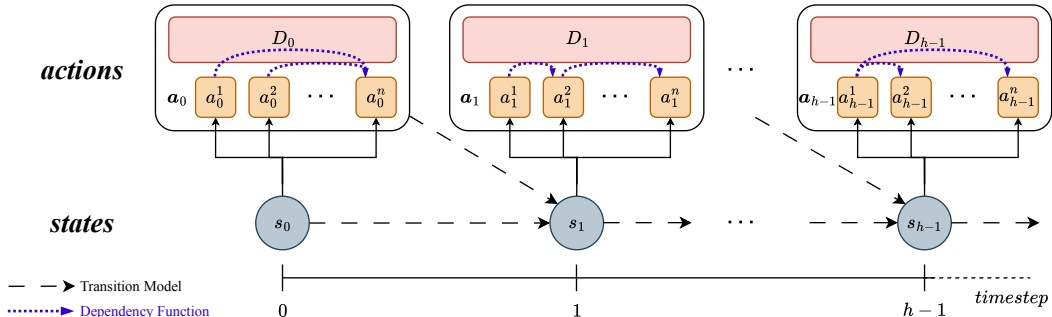

Figure 2: A detailed illustration of the dynamics of a Flex-MG. The dependency function (dashed purple line) can vary across timesteps. The superscript letters indicate agent index, from $1$ to $n$.

dynamic optimization demands of LaMAS. Subsequently, we transition from MARL to MARFT by introducing the Multi-Agent Advantage Decomposition Theorem and highlighting the key differences between MARFT and traditional MARL.

### 3.1 FLEXIBLE MARKOV GAME (FLEX-MG)

To enhance problem formulation and accommodate MARFT, we propose *Flexible Markov Game (Flex-MG)*, illustrated in Figure 2 and denoted as $\langle \mathcal{V}, \mathcal{N}, \mathcal{S}, \mathcal{A}, \mathcal{T}, \mathcal{R}, \gamma, \mathcal{D} \rangle$. Here, $\mathcal{V}$ is the vocabulary, $\mathcal{N} = \{1, ..., n\}$ is the agent composition of a LaMAS **with some organization**, i.e., some order constraint among agents, $\mathcal{S}$ is the state space, and $\mathcal{A} = \prod_{i=1}^{n} \mathcal{A}^i$ is the joint action space. The transition function $\mathcal{T}: \mathcal{S} \times \mathcal{A} \to \mathcal{S}'$, written as $P(s'|s, \boldsymbol{a})$, models state changes given joint actions. The reward function $\mathcal{R}: \mathcal{S} \times \mathcal{A} \to \mathbb{R}$, denoted $R(s, \boldsymbol{a})$, assigns rewards based on states and joint actions. The discount factor $\gamma$ accounts for temporal reward decay. Usually, the total reward at time $t$ is defined as $R_t \triangleq \sum_{k=0}^{\infty} \gamma^k r_{t+k}$.

Flex-MG is distinguished by its introduction of a **dependency function** $\mathcal{D}: \mathcal{A} \times \mathcal{A} \to \{0, 1\}$, which explicitly models inter-agent dependencies. Specifically, $\mathcal{D}(a^i, a^j) = 1$ indicates that the action of agent $j$ is conditionally dependent on the action of agent $i$. As a result, the decision-making process of agent $j$ is influenced not only by the global state $s$ but also by the actions of all agents $k$ such that $k \in \{l | \mathcal{D}(a^l, a^j) = 1\}$. These dependencies can be integrated into agent $j$'s input via concatenation or other fusion methods, thereby forming an enriched input for the agent's policy. Crucially, the dependency function $\mathcal{D}$ can vary across timesteps, potentially governed by an orchestration agent or a dynamic routing mechanism. This flexibility captures the evolving coordination structures in LaMAS and embodies the "flexible" nature of the proposed framework. Notably, when $\mathcal{D}(a^i, a^j) = 0$ for all $i, j$, Flex-MG reduces to a standard multi-agent decentralized setting where agents act independently.

### 3.2 TRANSITION FROM MARL TO MARFT

Before diving into MARFT, we introduce two critical value functions in MARL. The state value function $V^{\boldsymbol{\pi}}$ and the state-action value function $Q^{\boldsymbol{\pi}}$ are given by

$$V^{\boldsymbol{\pi}}(s) \triangleq \mathbb{E}_{\boldsymbol{a}_{0:\infty} \sim \boldsymbol{\pi}^{\boldsymbol{\theta}}, s_{1:\infty} \sim \mathcal{P}} [R_0 \mid s_0 = s], Q^{\boldsymbol{\pi}}(s, \boldsymbol{a}) \triangleq \mathbb{E}_{\boldsymbol{a}_{0:\infty} \sim \boldsymbol{\pi}^{\boldsymbol{\theta}}, \mathbf{s}_{1:\infty} \sim \mathcal{P}} [R_0 \mid s_0 = s, a_0 = \boldsymbol{a}].$$

The V-function indicates the value of being in state $s$, while the Q-function indicates how valuable the joint action $\boldsymbol{a}$ is at state $s$. Then, the advantage function is defined as $A^{\boldsymbol{\pi}}(s, \boldsymbol{a}) \triangleq Q^{\boldsymbol{\pi}}(s, \boldsymbol{a}) - V^{\boldsymbol{\pi}}(s)$, which quantifies how much better or worse the joint action $\boldsymbol{a}$ is compared to the average joint action at state $s$. Following these definitions, a key theorem enabling the transition to MARFT is the Multi-Agent Advantage Decomposition Theorem as shown in (1) below.

**Theorem 1** (Multi-Agent Advantage Decomposition Theorem (Kuba et al., 2021)). *For any predefined permutation of $n$ agents, for any state $s \in \mathcal{S}$ and joint action $\boldsymbol{a} \in \mathcal{A}$, the following always holds:*

$$A^{\boldsymbol{\pi}}(s, \boldsymbol{a}^{1:n}) = \sum_{m=1}^{n} A^{\boldsymbol{\pi}}(s, \boldsymbol{a}^{1:m-1}, \boldsymbol{a}^m), \tag{1}$$

This theorem reveals that if an agent is aware of the actions taken by its predecessors, maximizing the agent's local advantage is equivalent to maximizing the joint advantage. It provides a theoretical foundation to reformulate the problem as an SM task, similar to MAT (Wen et al., 2022), but MARFT aligns more naturally with the characteristics of LaMAS, making it a better-suited choice under dynamic organization.

Furthermore, in Table 1, we summarize the nuanced but critical distinctions between traditional MARL and MARFT, largely arising from LaMAS unique characteristics, such as action asynchronicity, agent profiles, and dynamic organizations etc. Detailed explanations are provided in Appendix C.

Table 1: Differences between MARFT and traditional MARL.

|  | **MARL** | **MARFT** |
| --- | --- | --- |
| Execution Asynchronicity | Synchronous | Asynchronous |
| Utilities | Task-specific only | Agentic with original language ability |
| Agent Identity | Unspecified, usually id | Specified profiles |
| Heterogeneity | Parameters | In-out formats, externals, parameters |
| System Organization | Static | Dynamic |
| Optimization Space | Small | Large and variable |

MARFT is designed to tackle the difficulties. To generally optimize the LaMAS regardless of the highly dynamic workflow or organization, MARFT reframes the problem as a sequential decision-making problem despite the dynamic organization. It preserves pretrained utilities by applying clipping, preventing excessive policy drift. Agent profiles are encoded to activate specific capabilities, and no modality assumptions are made, as all agent actions are expressed in tokens from their respective vocabularies.

## 4 MULTI-AGENT REINFORCEMENT FINE-TUNING

Building on the formulations outlined in the previous section, we now turn to the core methodologies that enable MARFT to address the unique challenges of optimizing LaMAS. Inheriting the SM-style re-modeling via the multi-agent advantage decomposition theorem introduced in the last section, MARFT follows the procedure demonstrated in Figure 3. Given any organizational or task-solving workflow of a LaMAS, theorem (1) allows MARFT to reframe it as a sequential decision-making process. When collecting interactive trajectories, the LaMAS generates actions with an auto-regressive problem-solving style. The "encoder" constructs local roll-out states using agent profiles and other relevant information, while the "decoder" combines these states with dependent predecessors' actions to generate the current agent's action $a^m \sim p_{\pi^m}(a^m|s, a^{1:m-1})$.

**MARFT-A** To instantiate the framework, we further present Action-level MARFT (MARFT-A), a variant designed for targeted, action-level policy optimization in LaMAS. During training, MARFT-A adopts an optimization objective in (2)[1]. Notably, in MARFT-A, each agent optimizes within a trust region conditioned on predecessor actions, akin to MAT, thus ensuring monotonic improvement as in HAPPO's sequential update scheme.

$$L(\boldsymbol{\theta}) = \frac{1}{nT} \sum_{i=1}^{n} \sum_{t=0}^{T-1} \min \left[ \boldsymbol{r}_t^m(\boldsymbol{\theta})\hat{A}_t, \text{clip}\left(\boldsymbol{r}_t^m(\boldsymbol{\theta}), 1 \pm \epsilon\right)\hat{A}_t \right], \quad (2)$$

$$\text{where } \boldsymbol{r}_t^m(\boldsymbol{\theta}) = \frac{\pi_{\theta^m}^m \left(a_t^m|s_t, \hat{\boldsymbol{a}}_t^{1:m-1}\right)}{\pi_{\theta_{\text{old}}^m}^m \left(a_t^m|s_t, \hat{\boldsymbol{a}}_t^{1:m-1}\right)}.$$

**MARFT-T** On top of the credit assignment of each agent on its action, we can also give a more fine-grained credit assignment by naturally treating every token as an action, leading to another token-level instantiation of MARFT named MARFT-T. In this situation, we need to define the token-level

---

[1]Our current MARFT instantiation employs a sequential decision-making process for scalability. Extending it to handle the parallel rollouts permitted by Flex-MG (i.e., a general DAG structure) would necessitate a more complex optimization objective to tackle off-policy problems, which we leave as a direction for future work.

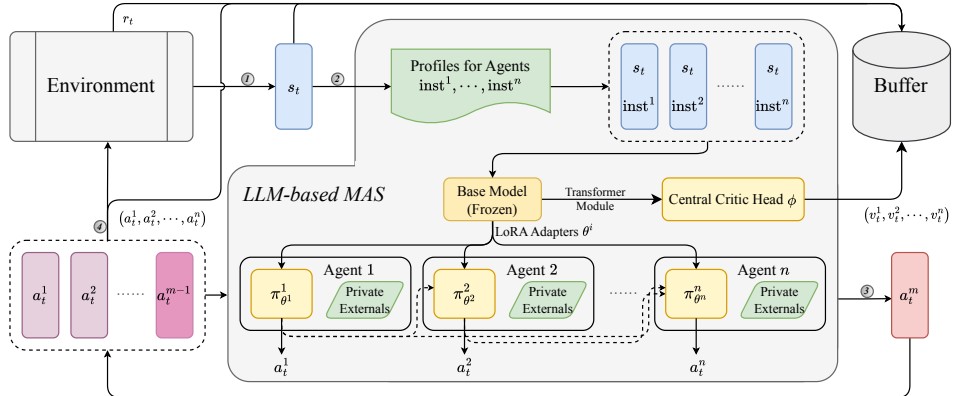

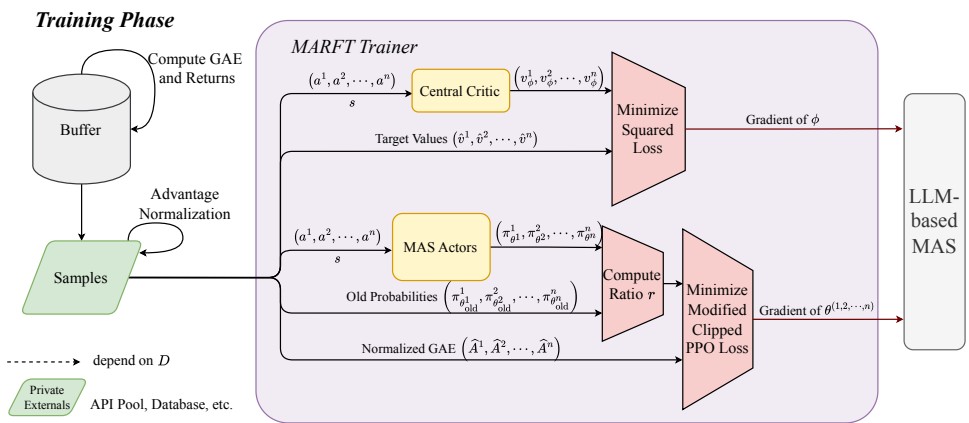

Figure 3: The procedure of Multi-Agent Reinforcement Fine-Tuning. Inference and training are conducted in an alternating manner. When the LaMAS is interacting with the environment and generating trajectories, the trajectory data will be stored in a ReplayBuffer. Then, during training, the data will be used to compute the values for optimization. The Central Critic Head is an extra trainable multilayer perceptron (MLP) with parameters $\phi$ to map the hidden vectors output by the frozen transformer module to specific values. Since each agent has its own role, when $\text{inst}^i$ is the agent's specific system prompt/profile.

multi-agent state value functions as

$$V^{\boldsymbol{\pi}}\left(s_t, \boldsymbol{a}_t^{i-1}, w_t^{i,j-1}\right) := \mathbb{E}_{\tau \sim p_{\boldsymbol{\pi}}\left(\tau | s_t, \boldsymbol{a}_t^{i-1}, w_t^{i,j-1}\right)} \left[\sum_{k=t}^{\infty} \gamma^{k-t} R_k(s_k, \boldsymbol{a}_k) | s_t, \boldsymbol{a}_t^{i-1}, w_t^{i,j-1}\right].$$

And then the token-level Bellman backup can be spontaneously derived as

$$V^{\boldsymbol{\pi}}(s_t, \boldsymbol{a}_t^{i-1}, w_t^{i,1:j}) \leftarrow \begin{cases} 0 + \gamma V^{\boldsymbol{\pi}}(s_t, \boldsymbol{a}_t^{1:i-1}, w_t^{i,1:j+1}) & \text{if } j < |a_t^i| \\ 0 + \gamma V^{\boldsymbol{\pi}}(s_t, \boldsymbol{a}_t^{1:i}, w_t^{i+1,1}) & \text{if } j = |a_t^i| \ \& \ i < n \ . \\ R(s_t, \boldsymbol{a}_t^{1:i}) + \gamma V^{\boldsymbol{\pi}}(s_{t+1}, \boldsymbol{a}_t^{1:i}, \emptyset) & \text{if } j = |a_t^i| \ \& \ i = n \end{cases}$$

This type of modified token-level Bellman backup guides the computations of TD errors and the advantages of tokens, thereby leading to token-level value training and policy optimization. In the context of the MARFT-T approach, each token is treated as an individual action, thereby introducing a new MG formulation, where the reward function is highly sparse and assigns zero rewards to all intermediate tokens and preceding agents, with the reward being allocated solely to the final token of the entire multi-agent system. Although this sparsity in the reward signal poses challenges for learning the value function, it significantly reduces the optimization complexity from $O\left(|\mathcal{V}|^L\right)$ to $O\left(|\mathcal{V}| \times L\right)$, which is discussed in Appendix C. However, due to the introduction of the different MG, MARFT-T optimizes inconsistently with the original optimization problem (Wen et al., 2024b).

# 5 EXPERIMENTS

To verify the effectiveness and superiority of the MARFT paradigm, we conduct a comprehensive evaluation. We first present the main results, comparing MARFT against the representative and state-of-the-art method MAPoRL (Park et al., 2025) (Independent PPO) to demonstrate its stability and performance advantages. Subsequently, we provide ablation studies and analysis, delving into the learning dynamics and the impact of different LaMAS organizations (e.g., SOLO, DUO, TRIO) to validate the internal mechanisms of MARFT. More experiment details are presented in Appendix D.

## 5.1 SETUPS

**Task Environments**  The environments are supported by specific datasets and reset by randomly sampling a (problem, answer, or test cases) pair. The LaMAS takes actions, and the environment verifier checks the correctness. The reward in the math problem-solving environment is 1 for correct answers and 0 otherwise, and the reward in the coding environment is the test case pass rate.

**Datasets and Benchmarks**  We use CodeForces (Penedo et al., 2025) for coding experiments. For math problem-solving, we use AIME 1983-2024 for the main comparison experiments, and MATH (Hendrycks et al., 2021) and CMATH (Wei et al., 2023) for detailed ablation and dynamics analysis.

**Base Model and LaMAS Setups**  We utilize the Qwen2.5 series (Yang et al., 2024a; Hui et al., 2024) as base models. In the main experiments, we train Qwen2.5-3B-Instruct on AIME and Qwen2.5-7B-Instruct on CodeForces, and in the ablation studies, We use Qwen2.5-Coder-3B-Instruct for math dynamics (MATH/CMATH) and Qwen2.5-3B-Instruct for coding dynamics. Additionally, we explore different LaMAS setups: SOLO (single-agent), DUO (Reasoner → Actor/Coder), and TRIO (Reasoner → Coder → Reviewer). Each agent is equipped with a dedicated LoRA adapter. Profile configurations are detailed in Appendix D.3.

## 5.2 COMPARISON WITH REPRESENTATIVE METHODS OF TUNING LAMAS

To demonstrate the superiority of MARFT, we compare it against MAPoRL (Park et al., 2025), a representative algorithm for tuning LaMAS. Since MAPoRL is fundamentally built upon Independent PPO (IPPO), where each agent maintains an individual critic and is trained independently, we adopt IPPO as the baseline.

**Baselines Setup**  Due to the inner nature of synchronous action in IPPO, a multi-agent debate-like LaMAS organization, i.e., each agent directly gives the answer, is used. Following MAPoRL's alignment, the reward for agent $i$ on math problems is $r^i = \frac{1}{2}\mathbb{I}(a^i = y) + \frac{1}{2n}\sum_{k=1}^{n}\mathbb{I}(a^k = y)$, and for coding problems is $r^i = \frac{1}{2}\text{PassRate}(a^i) + \frac{1}{2n}\sum_{k=1}^{n}\text{PassRate}(a^k)$, capturing both direct performance and team contribution.

**Training Dynamics**  Figure 4 illustrates the team reward curves during MARFT and IPPO training.

In the coding scenario, both methods initially show performance improvements. However, IPPO suffers from severe instability and catastrophic performance collapse after approximately 150 steps, indicated by the sharp drop in reward and exploding variance. In contrast, MARFT maintains a robust upward trend throughout the training process, demonstrating superior stability.

On the AIME environment, MARFT demonstrates consistent learning, contrasting with IPPO's stagnation. Although MARFT initially lags due to the temporary bottleneck of an unoptimized Reasoner, joint optimization enables rapid adaptation, allowing it to surpass IPPO after 300 steps and achieve superior convergence. These results empirically validate that MARFT's sequential decision-making pattern and multi-agent trust region learning effectively mitigate non-stationarity, whereas IPPO inherently lacks convergence guarantees (Tan, 1993).

**Evaluation Results**  Table 2 presents the quantitative evaluation results on the test sets. We observe that MARFT-A outperforms the MAPoRL (IPPO) baseline on both benchmarks. Specifically, on

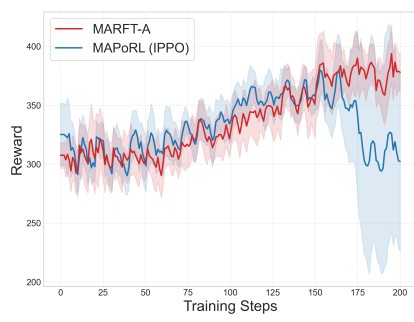
(a) Reward curves on CodeForces.

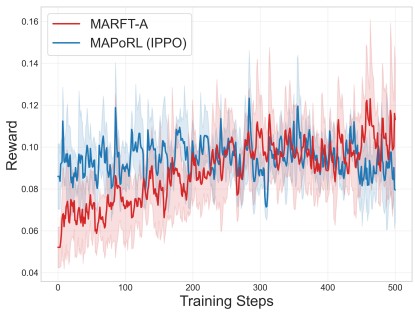
(b) Reward curves on AIME.

Figure 4: Reward curves of MARFT-A and MAPoRL (IPPO).

the CodeForces benchmark, MARFT-A achieves a score of 48.74, significantly surpassing IPPO by over 6 points. On the AIME benchmark, MARFT-A also demonstrates better performance (12.14% vs. 10.92%). Notably, MARFT-A exhibits a much lower standard error ($\pm 0.56$) compared to IPPO ($\pm 1.27$). These findings underscore MARFT-A's ability to achieve not only higher performance but also greater stability than the baseline.

Table 2: Evaluation results of MARFT-A and MAPoRL (IPPO).

| Benchmarks | MAPoRL (IPPO) | MARFT-A |
|---|---|---|
| CodeForces | $42.28 \pm 0.56$ | $\mathbf{48.74 \pm 1.05}$ |
| AIME | $10.92 \pm 1.27$ | $\mathbf{12.14 \pm 0.56}$ |

Furthermore, the results highlight the distinct suitability of different tasks for multi-agent systems. Mathematical reasoning often relies on atomic, linear logic that may not benefit from explicit decomposition, so we don't see much benefit coming out of multi-agent formulation. In contrast, coding inherently involves a clear division of labor (planning, coding, etc.) and a sequential decision-making process. This structural alignment makes coding a more compatible domain for the multi-agent paradigm than math, allowing MARFT to fully leverage its capabilities. Moreover, this preliminarily proves that MARFT is more scalable when applied to more complicatedly organized LaMAS for more agentic tasks.

## 5.3 ABLATION STUDIES AND ANALYSIS

Having established the superiority of MARFT over independent training methods, we now analyze the internal learning dynamics and the impact of different LaMAS configurations, and also conduct an ablation experiment on optimization granularity.

### 5.3.1 LEARNING DYNAMICS

Figure 5 shows the training dynamics of MARFT-A of DUO in environments supported by MATH and CMATH. The episodic return (ER) is the correctness of solving the math problem. The average step reward (ASR) is calculated by dividing the episodic return by the steps taken to solve the problem.

We observe that in math problem-solving environments, the episodic return exhibits a clear improvement up to about 8 p.p.[2] (~18.45%). However, during the early stages of training, the episodic return demonstrates noticeable oscillations. This instability can be explained by the behavior of the value loss, which display high losses during the initial phase. Besides, we observe a constant improvement in average step reward, which means DUO not only tries to achieve high episodic return but also learns to solve the problem more efficiently.

Figure 6 shows the training dynamics of MARFT-A of DUO and TRIO in coding environments built by CodeForces. We observe that the scores undergo a certain oscillation in the beginning and then increase stably from about 220 to about 260 (~18.18%) and 280 (~27.27%) respectively for DUO and TRIO.

---
[2]p.p. is the abbreviation for percentage points.

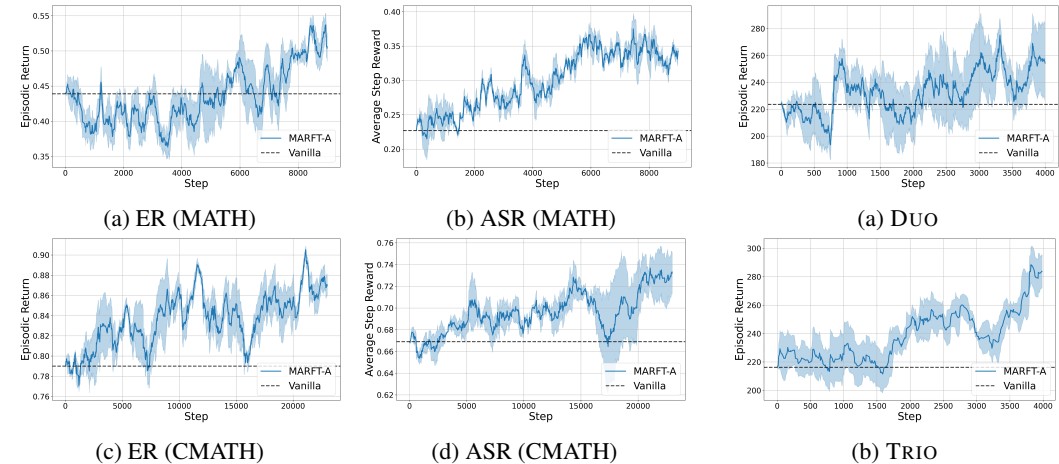

(a) ER (MATH)                (b) ASR (MATH)                (a) Duo

(c) ER (CMATH)              (d) ASR (CMATH)              (b) Trio

Figure 5: Learning dynamics of MARFT-A of Duo on MATH and CMATH.

Figure 6: Learning dynamics on CodeForces.

### 5.3.2 EVALUATION RESULTS ANALYSIS

On top of the learning dynamics of MARFT-A in the environment supported by the train sets, we also conduct evaluations on the test sets. Table 3 presents the performance of SOLO, DUO, and MARFT-A-tuned DUO on the MATH500, CMATH, and GSM8K test sets, including in-domain and out-of-domain assessments. And Table 4 gives a comparison of total scores between vanilla LaMAS and the MARFT-A-tuned ones on the test set.

Table 3: Evaluation results (in percentages) of SOLO, DUO and MARFT-A-tuned DUO on mathematical benchmarks. MARFT-A DUO-MATH and MARFT-A DUO-CMATH mean MARFT-A-tuned DUO in the environments supported by MATH train set and CMATH train set.

| Benchmarks | SOLO | DUO | MARFT-A | |
| --- | --- | --- | --- | --- |
| | | | DUO-MATH | DUO-CMATH |
| MATH500 | $36.44 \pm 1.35$ | $43.9 \pm 1.09$ | $47.14 \pm 1.31$ | $\mathbf{47.18 \pm 0.10}$ |
| CMATH | $80.24 \pm 0.62$ | $79.00 \pm 0.98$ | $81.26 \pm 0.55$ | $\mathbf{81.82 \pm 0.85}$ |
| GSM8K | $68.45 \pm 1.68$ | $74.23 \pm 0.63$ | $\mathbf{77.24 \pm 0.94}$ | $76.76 \pm 0.55$ |

**SOLO vs. Multi-Agent** For mathematical tasks, by comparing SOLO and DUO, we observe that DUO significantly outperforms SOLO on the MATH500 benchmark by approximately 7.5 p.p. (~20.58%). However, their performance is comparable on CMATH and GSM8K, likely because these datasets consist of relatively simpler problems. In such cases, a more complex solving process may hinder performance. These results suggest that a LaMAS like DUO is more effective than a single agent in solving complex mathematical problems. For coding tasks, SOLO and DUO perform close to each other, but TRIO falls behind by about one point, which proves that overcomplicating a system for a task with modest difficulty can harm the system's performance.

**Vanilla vs. MARFT-A** In mathematical tasks, MARFT-A improves DUO by ~3 p.p. in-domain while demonstrating strong cross-lingual generalization: DUO-CMATH scores 47.18% on MATH500, and DUO-MATH reaches 81.26% on CMATH. Consistently improved performance on GSM8K further confirms MARFT's ability to foster robust generalization. In coding, MARFT-A boosts DUO by ~4 points (~14.75%), outperforming the tuned SOLO by ~2.5 points. Even for the suboptimal TRIO setup, MARFT-A yields a ~2-point gain (~6.81%), validating its effectiveness regardless of architectural complexity.

Table 4: Evaluation results of SOLO, DUO, TRIO, and the MARFT-A-tuned ones on CodeForces.

| Setup | Vanilla | MARFT-A |
| --- | --- | --- |
| SOLO | $27.21 \pm 2.84$ | $28.92 \pm 2.07$ |
| DUO | $27.45 \pm 1.60$ | $\mathbf{31.50 \pm 1.16}$ |
| TRIO | $26.58 \pm 2.47$ | $28.39 \pm 1.58$ |

### 5.3.3 OPTIMIZATION GRANULARITY

Figure 7 (Supplementary Figure 8 in Appendix) shows the performance of MARFT-A and MARFT-T while fine-tuning DUO, in mathematical environments. From both subfigures, we see that MARFT-A is overall better than MARFT-T. Specifically, MARFT-A achieves a higher episodic return than MARFT-T in the end and has a more stable and faster improvement. This kind of superiority of MARFT-A over MARFT-T can be theoretically deduced from the inconsistency of their optimality, which is caused by the different MG (Wen et al., 2024b).

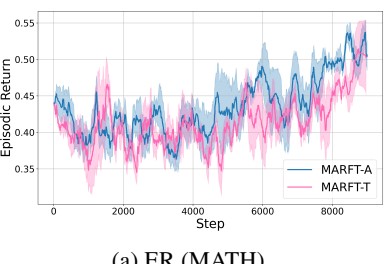 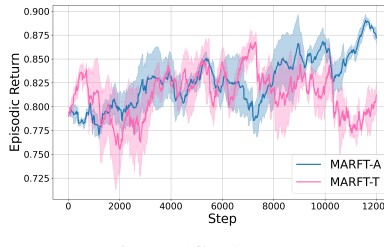

(a) ER (MATH).  (b) ER (CMATH).

Figure 7: Curves of episodic return of MARFT-A and MARFT-T while fine-tuning DUO.

## 6 CONCLUSION

In this work, we propose MARFT, a groundbreaking paradigm that pushes the extension of MARL principles to general LaMAS. Unlike prior approaches that rely on static supervision or homogeneous agent design, MARFT treats LaMAS as a dynamic, heterogeneous collective that learns through experience-driven interaction. By formalizing the Flex-MG and employing a sequential optimization paradigm rooted in the Multi-Agent Advantage Decomposition Theorem, MARFT enables scalable, fine-grained reinforcement fine-tuning in complex, real-world agentic environments. Empirically, comparative analyses against representative methods like MAPoRL (IPPO) highlight MARFT's superior stability and performance, validating its robustness over independent training methods. Furthermore, extensive experiments on mathematical and coding tasks demonstrate the general effectiveness of MARFT's instantiated algorithms and the impact of LaMAS configurations. This work truly represents a significant step toward aligning RFT with the nuanced architecture of LaMAS, paving the way for more adaptive, collaborative, and autonomous LLM agent ecosystems.

## 7 LIMITATIONS AND FUTURE WORK

**Limitations** Despite the advantages and potential of MARFT, both academia and industry face significant challenges that hinder the development of more effective MARFT algorithms. First, there is an urgent need for a comprehensive, real-world-like multi-agent benchmark. Existing mathematical and coding scenarios are relatively narrow in scope, often allowing a single competent agent to solve tasks independently, which diminishes the relevance of LaMAS and MARFT in such settings. More complex environments tailored for LaMAS with truly agentic and collaborative tasks are essential, yet currently absent. Moreover, large-scale MARFT demands rapid and parallel rollout mechanisms to gather sufficient trajectories for optimization. A lack of sufficient trajectories can lead to high variance and instability during training, ultimately resulting in slow convergence. A complete, efficient, and effective codebase for MARFT is urgently needed to extensively scale up environments, agent population, model size, etc.

**Future Work** In light of these limitations and challenges, future research should focus on developing LaMAS-oriented benchmarks that emulate real-world environments and thoroughly evaluate the agentic capabilities of LaMAS. Moreover, optimizing the existing codebase and framework to support large-scale training is essential. This would further enable more effective MARFT implementation, e.g., by fine-tuning agents for complex, mainstream agentic workflows like CAMEL-OWL. Handling such workflows, which often feature parallel interactions better represented as Directed Acyclic Graphs (DAGs), would necessitate an evolution of the current algorithm, marking a promising avenue for future work. This approach is feasible as the MARFT paradigm is decoupled from any specific LaMAS architecture. Such optimizations would also significantly accelerate progress in conducting experiments with more agents to discover a potential Agent Scaling Law.

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

APPENDIX

## A   THE USE OF LARGE LANGUAGE MODELS

We employed large language models (LLMs) solely to polish the writing of this paper, such as improving grammar, clarity, and readability. The models were not used for generating original ideas, experiments, analyses, or results. All scientific contributions, methods, and conclusions presented in the paper are entirely the work of the authors.

## B   MARFT-A ALGORITHM DETAILS

---

**Algorithm 1:** Action-level Multi-Agent Reinforcement Fine-Tuning (MARFT-A)

---

**Input:** Agent population $n$, agent profiles $\text{inst}^i$, the initial joint policy $\boldsymbol{\pi}_{\boldsymbol{\theta}_0}$ with parameters $\theta_0^i$ for each policy $\pi_{\theta^i}^i$, initial parameters $\phi_0$ for the critic network $V_\phi$, hyper-parameters including maximum interaction steps $T$ in one roll-out, clip parameter $\epsilon$, discount factor $\gamma$, GAE $\lambda$, etc.

**Output:** Optimized joint policy $\boldsymbol{\pi}$ and critic network $\phi$

1  initialize policy $\boldsymbol{\pi}_{\boldsymbol{\theta}} \leftarrow \boldsymbol{\pi}_{\boldsymbol{\theta}_0}$, critic network $V_\phi \leftarrow V_{\phi_0}$ and buffer $\mathcal{D} \leftarrow \emptyset$

2  **for** *episode* = 0, . . . **do**

3     **for** $t = 0, \ldots, T - 1$ **do**

4        collect $s_t$

5        **for** $i = 1, \ldots, n$ **do**

6           generate action $a^i \sim p_{\pi_{\theta^i}^i}(a^i | s, \boldsymbol{a}^{1:i-1})$

7        **end**

8        $s_{t+1} \sim \mathcal{T}(\cdot | s_t, \boldsymbol{a}_t)$

9        $r_t = R(s_t, \boldsymbol{a}_t)$

10       $\mathcal{D} \leftarrow \mathcal{D} \cup \{(s_t, \boldsymbol{a}_t, r_t, s_{t+1})\}$

11    **end**

12    compute advantage estimate $\hat{A}$ via GAE and compute value network target $\hat{V}$ in $\mathcal{D}$

13    **for** *n epoches* **do**

14       sample a batch $\mathcal{B} = \{(s_t, \boldsymbol{a}_t, r_t, s_{t+1}, \hat{A}, \hat{V})\} \sim \mathcal{D}$

15       update $\phi$ by minimizing $\sum_{n=1}^{N} \left\| V_\phi(s_n) - \hat{V}_n \right\|^2$

16       update $\boldsymbol{\theta}$ by maximizing the objective (2)

17    **end**

18 **end**

---

## C   DETAILED EXPLANATIONS OF THE DIFFERENCES BETWEEN MARL AND MARFT

**Execution Asynchronicity.** One of the most significant differences between conventional MARL and MARFT is the nature of agent actions. In traditional cooperative MARL, agents typically act simultaneously. However, in MARFT, actions of agents often execute asynchronously. In some cases, the actions of certain agents may even depend on the outcomes of other agents' actions' execution. For example, in a collaborative coding assistant system, one LLM agent generates a function prototype, and another agent asynchronously refines it based on the first agent's output, demonstrating both asynchronicity and result dependency. As a result, conventional MARL methods, which assume synchronous actions, may not be directly applicable or effective in LLM-based multi-agent systems.

**Utilities.** Unlike agents in typical MARL problems, LLMs are initially designed for language processing rather than specific agentic tasks. However, the value- or reward-guided nature of RL means that when using value-based RL algorithms to optimize or extract policies, the derived optimal policy often prioritizes actions that maximize the value function. This can lead to policies that generate high-reward actions or tokens but are incomprehensible to humans. Though some attempts to extract a policy together with entropy regularization have been made to improve its agentic intelligence without harming the text capability (Wen et al., 2024a), it is still a point that requires additional attention when we try to implement value-based methods to optimize LLMs.

**Characteristic Profiles.** The transition from single-agent to multi-agent systems in typical MARL benchmarks, such as Multi-Agent MuJoCo, often involves splitting a single agent into multiple components without additional modifications. In contrast, LLMs require a more nuanced approach. When decomposing tasks into orthogonal sub-tasks, each LLM agent needs a profile to define its role and capabilities, enabling it to generate actions consistent with its assigned character. This profile can be human-specified or learned by the agent through natural evolution. Consequently, the joint observation space in MARFT is augmented with profiles, taking the form *[agent profile + environment state (+ agent-specific instruction)]*. This modification is crucial when designing a MARL training framework.

**Heterogeneity.** In conventional MARL benchmarks, heterogeneity is typically characterized by differences in agent structures and non-parameter-sharing schemes. However, LLM-based systems introduce a higher level of complexity. First, LLMs themselves can vary significantly in terms of model structures, parameters, input and output formats (e.g., LLMs vs. Vision-Language Models), and vocabularies. Second, agents may have access to different external tools or devices, reflecting real-world scenarios. For instance, in designing a multi-agent system for a mobile operating system, some agents may have access to both local and remote search engines, while others may rely on proprietary databases or tools from other companies. This heterogeneity complicates training, making it more difficult and unstable.

**System Organization.** Traditional MARL tasks are often set in static simulated environments. In contrast, LLM-based multi-agent systems are designed for real-life agentic tasks with higher uncertainty. These tasks can be decomposed in various ways, leading to different multi-agent systems with distinct populations and roles. Moreover, agents may exhibit sequential dependencies and contextual relationships when solving sub-tasks. For example, one agent's action may depend on the outcome of another agent's action. This dynamic organization can be either human-designed or agent-explored (e.g., through training), adding another layer of complexity to the design process.

**Optimization Space.** If we take one LLM generation as an action, the observation space is exponentially vast as $o \in \mathcal{O} \subseteq \mathcal{V}^L$, whose complexity is $O(|\mathcal{V}|^L)$, where $\mathcal{V}$ is the vocabulary of the acting LLM and $N$ is the token length of the generated action, and both of them vary with different acting LLMs. It makes the value function hard to converge with stability. If we take one token as an action, the complexity can be reduced to $O(|\mathcal{V}| \times L)$, but it will become a task with super sparse reward outcome so that the value function should also be meticulously learned, as $L$ can be extremely large. This is also reflected in VAPO and is mitigated by value pretraining (Yue et al., 2025). Fortunately, not all LLM-related tasks face such large optimization spaces. For instance, in embodied AI with LLMs, predefined actions (e.g., "go to the kitchen," "grab a coffee") are often provided, significantly narrowing the action space (Carta et al., 2023; Tan et al., 2024).

# D EXPERIMENT DETAILS

## D.1 MORE EXPERIMENT CURVES

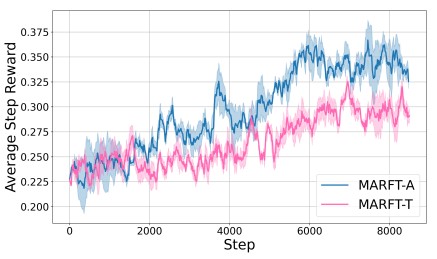

(a) Average Step Reward (MATH).

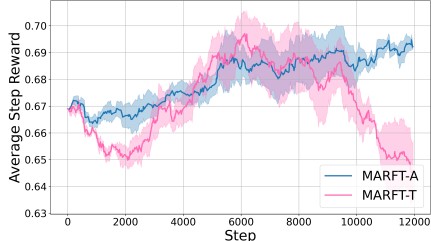

(b) Average Step Reward (CMATH).

Figure 8: Curves of average step reward of MARFT-A and MARFT-T while fine-tuning DUO.

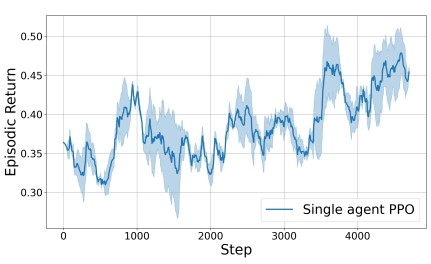

(a) Episodic Return (MATH).

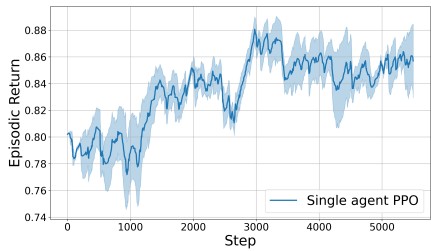

(b) Episodic Return (CMATH).

Figure 9: Curves of episodic return during SOLO MARFT.

## D.2 HYPERPARAMETER CONFIGURATIONS FOR EXPERIMENTS

Table 5: Hyperparameter configuration for learning dynamics in Figure 5.

| Hyperparameter | Value | Hyperparameter | Value | Hyperparameter | Value |
|---|---|---|---|---|---|
| MAS learning rate | 1e-6 | critic learning rate | 1e-5 | use gae | True |
| rollout threads | 1 | num mini-batch | 1 | ppo epoch | 1 |
| max horizon | 2 | hidden size | 64 | episode length | 8 |
| optim eps | 1e-5 | critic loss coef | 1 | entropy coef | 1e-3 |
| gamma | 0.99 | gae gamma | 0.95 | max grad norm | 0.5 |
| context window | 2456 | max new tokens | 512 | clip param | 0.2 |

Table 6: Hyperparameter configuration for learning dynamics in Figure 6.

| Hyperparameter | Value | Hyperparameter | Value | Hyperparameter | Value |
|---|---|---|---|---|---|
| MAS learning rate | 1e-6 | critic learning rate | 1e-5 | use gae | True |
| rollout threads | 4 | num mini-batch | 1 | ppo epoch | 1 |
| max horizon | 1 | hidden size | 64 | episode length | 30 |
| optim eps | 1e-5 | critic loss coef | 1 | entropy coef | 1e-3 |
| gamma | 0.99 | gae gamma | 0.95 | max grad norm | 0.5 |
| context window | 4096 | max new tokens | 1024 | clip param | 0.2 |

Table 7: Hyperparameter configuration for MARFT-A in Figure 4.

| Hyperparameter | Value | Hyperparameter | Value | Hyperparameter | Value |
|---|---|---|---|---|---|
| MAS learning rate | 3e-5 | critic learning rate | 3e-5 | use gae | True |
| rollout threads | 128 | num mini-batch | 1 | ppo epoch | 1 |
| optim eps | 1e-8 | max new tokens | 2048 | share critic | True |
| gamma | 1.0 | context window | 32768 | clip param | 0.2 |

Table 8: Hyperparameter configuration for IPPO in Figure 4.

| Hyperparameter | Value | Hyperparameter | Value | Hyperparameter | Value |
|---|---|---|---|---|---|
| MAS learning rate | 3e-5 | critic learning rate | 3e-5 | use gae | True |
| rollout threads | 128 | num mini-batch | 1 | ppo epoch | 1 |
| optim eps | 1e-8 | max new tokens | 2048 | share critic | False |
| gamma | 1.0 | context window | 32768 | clip param | 0.2 |

Table 9: Hyperparameter configuration for MARFT-T in Figure 7.

| Hyperparameter | Value | Hyperparameter | Value | Hyperparameter | Value |
|---|---|---|---|---|---|
| MAS learning rate | 3e-6 | critic learning rate | 1e-5 | use gae | True |
| rollout threads | 1 | num mini-batch | 1 | ppo epoch | 1 |
| max horizon | 2 | hidden size | 64 | episode length | 8 |
| optim eps | 1e-5 | critic loss coef | 1 | entropy coef | 1e-3 |
| gamma | 0.99 | gae gamma | 0.95 | max grad norm | 0.5 |
| context window | 2456 | max new tokens | 512 | clip param | 0.2 |

Table 10: Hyperparameter configuration for fine-tuning SOLO in Figure 9.

| Hyperparameter | Value | Hyperparameter | Value | Hyperparameter | Value |
|---|---|---|---|---|---|
| MAS learning rate | 3e-6 | critic learning rate | 1e-5 | use gae | True |
| rollout threads | 2 | num mini-batch | 1 | ppo epoch | 1 |
| max horizon | 2 | hidden size | 64 | episode length | 8 |
| optim eps | 1e-5 | critic loss coef | 1 | entropy coef | 1e-3 |
| gamma | 0.99 | gae gamma | 0.95 | max grad norm | 0.5 |
| context window | 1536 | max new tokens | 512 | clip param | 0.2 |

### D.3 PROFILES

For the profiles that define distinct agent characteristics, we have designed two configurations for math problems and three configurations for coding problems as below:

For math problem-solving scenario:

SOLO (Single-agent) profile:

```
[
    {
        "role": "actor",
        "prompt": "<|im_start|>system: You are the **Actor**, an LLM
            agent responsible for solving math problems. Analyze the
            problem, determine the optimal solution path, execute
            calculations, and provide the final answer within \\boxed
            {{}}.<|im_end|>\n",
        "with_answer": true
    }
]
```

DUO (Duo LaMAS) profiles:

```
[
    {
        "role": "reasoner",
        "prompt": "<|im_start|>system: Two LLM agents (Reasoner -> Actor)
            collaborate step-by-step to solve math problems. You are the
            **Reasoner**: Analyze the original problem, historical
            actions, and reflection data (if provided) to determine the
            critical next step. Guide the Actor by providing concise
            reasoning for the optimal operation.<|im_end|>\n",
        "with_answer": false
    },
    {
        "role": "actor",
        "prompt": "<|im_start|>system: Two LLM agents (Reasoner -> Actor)
            collaborate step-by-step. You are the **Actor**: Execute
            operations using the original problem, action history, and
            Reasoner's guidance. Provide final answer within \\boxed
            {{}}.<|im_end|>\n",
```

```
10        "with_answer": true
11    }
12 ]
```

For coding scenario:

SOLO (Single-agent) profile:

```
1  [
2      {
3          "role": "coder",
4          "prompt": "<|im_start|>You are the **Coder**: Solve the coding
               problem using efficient, correct, and clean Python code.
               Handle edge cases, respect problem constraints, and structure
                your solution for clarity.\nAlways use Python.<|im_end|>",
5          "with_answer": true,
6      }
7  ]
```

DUO (Duo LaMAS) profiles:

```
1  [
2      {
3          "role": "reasoner",
4          "prompt": "<|im_start|>system: Two LLM agents (Reasoner -> Coder)
                collaborate to solve Codeforces Python coding problems. You
               are the **Reasoner**: Analyze the problem statement,
               constraints, and expected behavior. Identify edge cases,
               break the problem into logical steps, and suggest an
               algorithmic plan. You may include helpful pseudocode, but do
               **not** write actual Python code.<|im_end|>",
5          "with_answer": false,
6      },
7      {
8          "role": "coder",
9          "prompt": "<|im_start|>system: Two LLM agents (Reasoner -> Coder)
                collaborate to solve Codeforces problems using Python. You
               are the **Coder**: Implement the Reasoner's plan using
               efficient, correct, and clean Python code. Handle edge cases,
                respect problem constraints, and structure your solution for
                clarity.\nAlways use Python.<|im_end|>",
10         "with_answer": true,
11     }
12 ]
```

TRIO (Trio LaMAS) profiles:

```
1  [
2      {
3          "role": "reasoner",
4          "prompt": "<|im_start|>system: Three LLM agents (Reasoner ->
               Coder -> Reviewer) collaborate to solve Codeforces Python
               coding problems.\nYou are the **Reasoner**: Analyze the
               problem statement, constraints, and expected behavior.\
               nIdentify edge cases, break the problem into logical steps,
               and suggest a high-level algorithmic plan.\nYou may include
               helpful pseudocode and edge case analysis, but do **not**
               write actual Python code.<|im_end|>",
5          "with_answer": false,
6      },
7      {
8          "role": "coder",
9          "prompt": "<|im_start|>system: Three LLM agents (Reasoner ->
               Coder -> Reviewer) collaborate to solve Codeforces Python
```

```
                coding problems.\nYou are the **Coder**: Implement the
                Reasoner's plan using efficient and correct Python code.\
                nHandle edge cases, follow the provided strategy, and ensure
                clarity and correctness.\nAlways use Python.\nPlace your
                complete solution below the line starting with 'Answer:'.<|
                im_end|>",
10          "with_answer": false,
11      },
12      {
13          "role": "reviewer",
14          "prompt": "<|im_start|>system: Three LLM agents (Reasoner ->
                Coder -> Reviewer) collaborate to solve Codeforces Python
                coding problems.\nYou are the **Reviewer**: Carefully review
                the Coder's solution.\n1. Check for correctness against the
                problem and Reasoner's plan.\n2. Identify and add any missing
                 test cases, especially edge cases.\n3. Suggest or perform
                optimizations if needed (e.g. time or memory improvements).\
                n4. Provide a final, polished version of the code if
                improvements are made.\nStart your message with 'Review:',
                and include improved Python code (if any) below 'Final Answer
                :'.<|im_end|>",
15          "with_answer": true,
16      }
17  ]
```

# E  EXTENDED LEARNING DYNAMICS AND EMERGENT BEHAVIORS

## E.1  ACCURACY CURVE OF REASONER WHILE FINE-TUNING

We observe an intriguing emergent behavior when training the DUO setup on AIME. Figure 10 tracks the accuracy of the *Reasoner*. Although the system prompt explicitly constrains it to "only provide high-level plans and do not directly provide answers", the agent learns to bypass this instruction and generate the final answer directly to maximize the team reward. This phenomenon underscores a fundamental characteristic of RL: optimization is driven strictly by the reward signal, which can often compromise instruction-following capabilities if deviating from the prompt yields a higher return.

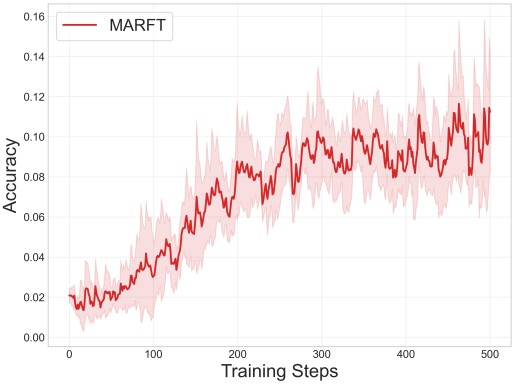

Figure 10: Curve of the accuracy of Reasoner in DUO while being trained using MARFT on AIME.

## E.2  CASE STUDIES AND EMERGENT BEHAVIORS

```
user
Execution time limit: 1.0 seconds
Memory limit: 256.0 MB

# Problem
```

Today, Sakurako has a math exam. The teacher gave the array, consisting of $a$ ones and $b$ twos.

In an array, Sakurako must place either a '+' or a '-' in front of each element so that the sum of all elements in the array equals $0$.

Sakurako is not sure if it is possible to solve this problem, so determine whether there is a way to assign signs such that the sum of all elements in the array equals $0$.

## Input Format

The first line contains a single integer $t$ ($1\le t\le 100$) -- the number of test cases.

The only line of each test case contains two integers $a$ and $b$ ($0\le a, b<10$) -- the number of '1's and the number of '2's in the array.

## Output Format

For each test case, output "Yes" if you can make the sum of the entire array equal to $0$, and "No" otherwise.

You can output each letter in any case (lowercase or uppercase). For example, the strings "yEs", "yes", "Yes", and "YES" will be accepted as a positive answer.

## Examples

```input
5
0 1
0 3
2 0
2 3
3 1
```

```output
NO
NO
YES
YES
NO
```

## Note
1. $a=0$, $b=1$: This means the array is $[2]$ -- it is impossible to add the signs '+' or '-' to get $0$ as a result;
2. $a=0$, $b=3$: This means the array is $[2, 2, 2]$ -- it is impossible to add the signs '+' or '-' to get $0$ as a result;
3. $a=2$, $b=0$: This means the array is $[1, 1]$ -- it is possible to add the signs '+' or '-' to get $0$ as a result ($+1-1=0$);
4. $a=2$, $b=3$: This means the array is $[1, 1, 2, 2, 2]$ -- it is possible to add the signs '+' or '-' to get $0$ as a result ($+1+1-2-2+2=0$);

Reasoner: ### High-Level Architectural Plan

1. **Input Reading:**

- Read the number of test cases, `t`.
- For each test case, read the values of `a` (number of '1's) and `b` (number of '2's).

2. **Check Feasibility:**
   - Calculate the total sum of the array without any signs. This is given by `3 * b + a`.
   - Determine if this total sum can be split into two equal parts (i.e., if the total sum is even and `b` is not too high to make the sum zero).

3. **Conditions for Zero Sum:**
   - If the total sum is odd, it is impossible to make the sum zero. Output "NO".
   - If the total sum is even, check the following conditions:
     - If `b` is greater than `a`, it is impossible to make the sum zero. Output "NO".
     - If `b` is less than or equal to `a`, it is possible to make the sum zero. Output "YES".

4. **Output:**
   - For each test case, print "YES" if the sum can be zero, otherwise print "NO".

### Pseudocode
```plaintext

// Main Function
for each test case
    read a, b
    calculate total_sum = 3 * b + a
    if total_sum is odd
        output "NO"
    else if b > a
        output "NO"
    else
        output "YES"
```

### Detailed Steps

1. **Read Input:**
   - `t = read_integer()`
   - for `i` from 1 to `t`
     - `a, b = read_integers()`

2. **Calculate Total Sum:**
   - `total_sum = 3 * b + a`

3. **Check Total Sum:**
   - `if total_sum % 2 != 0`
     - `output "NO"`
   - `else if b > a`
     - `output "NO"`
   - `else`
     - `output "YES"`

This plan outlines the logic and control flow needed to solve the problem efficiently within the given constraints.

Coder: ```python

```python
# Define the main function to handle the logic
def main():
    # Read the number of test cases
    t = int(input())
    # Process each test case
    for _ in range(t):
        # Read the values of a and b
        a, b = map(int, input().split())
        # Calculate the total sum
        total_sum = 3 * b + a
        # Check if the total sum can be zero
        if total_sum % 2 != 0:
            print("NO")
        elif b > a:
            print("NO")
        else:
            print("YES")

# Call the main function
if __name__ == "__main__":
    main()
```

This code follows the high-level plan provided by the reasoner:
1. It reads the number of test cases.
2. For each test case, it reads the values of `a` and `b`.
3. It calculates the total sum of the array without any signs.
4. It checks if the total sum can be split into two equal parts and prints "YES" or "NO" accordingly.
5. The main function is called to execute the logic.

