# OpenReview forum: "MARFT: Multi-Agent Reinforcement Fine-Tuning"
_ICLR.cc/2026/Conference — Submitted to ICLR 2026_

### Official Review · Reviewer_xPkj · 2025-10-19

**Soundness:** 2
**Presentation:** 2
**Contribution:** 1
**Rating:** 2
**Confidence:** 4

**Summary:**

This paper introduces **MARFT (Multi-Agent Reinforcement Fine-Tuning)**, a framework that aims to extend reinforcement learning (RL) principles to **LLM-based multi-agent systems (LaMAS)**. The authors first formalize a new game model, the **Flexible Markov Game (Flex-MG)**, which introduces a dynamic dependency function ( D(a_i, a_j) ) to capture conditional relations between agents. Building on this formulation, they propose two concrete implementations:

* **MARFT-A** (action-level fine-tuning), where each agent’s policy is optimized with a PPO-like objective under sequential dependencies;
* **MARFT-T** (token-level fine-tuning), which treats every token as an action and defines a token-level Bellman backup.

The authors evaluate MARFT on **mathematical problem-solving** (MATH, CMATH, GSM8K) and **coding** (CodeForces) environments using Qwen2.5 models in single-, dual-, and triple-agent configurations. Experimental results show modest accuracy improvements (e.g., +3 p.p. on MATH500 and +4 points on CodeForces) compared with vanilla multi-agent or single-agent baselines.

The paper positions MARFT as a new paradigm that unifies large-language-model fine-tuning and multi-agent reinforcement learning by leveraging heterogeneous, dynamically organized agents interacting through language.

**Strengths:**

The paper addresses a timely problem, optimizing large language model-based multi-agent systems (LaMAS) through reinforcement learning, and presents a reasonably clear exposition of the proposed framework. The idea of introducing a Flexible Markov Game (Flex-MG) to capture dynamic dependencies among agents is conceptually interesting, and the implementation of two variants (MARFT-A and MARFT-T) demonstrates some engineering effort and reproducibility. The manuscript is well written and organized, with clear figures and experimental details that make the pipeline understandable. Overall, the paper is sound at a conceptual level and fair in its presentation quality, showing adequate awareness of related work and providing a structured attempt to formalize multi-agent fine-tuning for LLMs, even though the originality and practical contribution remain limited.

**Weaknesses:**

The main weakness of this paper lies in its **lack of conceptual clarity and empirical justification**. Although the Flex-MG formulation introduces a dependency function $D(a_i, a_j)$, the paper does not clearly demonstrate how these dependencies are represented or learned in practice. No concrete examples or visualizations in experiment part are provided to help readers understand how one agent’s action influences another, leaving the proposed mechanism largely theoretical and disconnected from real implementation. The **experimental design is weak**, as all evaluations are conducted on mathematical and coding tasks that do not inherently require multi-agent coordination. The “Reasoner–Actor” and “Coder–Reviewer” setups appear artificial and fail to convincingly illustrate any genuine inter-agent interaction or dependency. Moreover, the **performance improvements are minor** and could easily stem from additional fine-tuning rather than from the multi-agent reinforcement framework itself. The paper also **fails to position MARFT relative to RLHF**: it uses standard PPO objectives without explaining the conceptual or methodological distinction from human-feedback-based reinforcement learning, which is currently the dominant paradigm for post-training LLMs. Overall, the work feels **premature and speculative**, with limited novelty beyond reinterpreting existing MARL ideas under LLM settings and insufficient empirical evidence to support its claimed contributions.

**Questions:**

1. **Clarification on action dependency modeling:** Could the authors provide a concrete example or visualization showing how the dependency function ( D(a_i, a_j) ) is computed or updated during training? For instance, how does one agent’s output affect another’s input in the MARFT framework, and how is this handled across asynchronous timesteps?

2. **Justification for multi-agent design in math and coding tasks:** Why are tasks like MATH and CodeForces appropriate for evaluating multi-agent reinforcement learning? Can the authors demonstrate any scenario where coordination between agents (e.g., Reasoner–Actor or Coder–Reviewer) is essential to solving the task, rather than just increasing model complexity?

3. **Connection and difference with RLHF:** The paper repeatedly refers to “reinforcement fine-tuning,” but appears to use standard PPO without human feedback. Could the authors clearly articulate how MARFT differs from RLHF in terms of training signal, objective function, or supervision source?

4. **Empirical evidence of coordination benefits:** Beyond modest accuracy gains, is there any qualitative or behavioral analysis (e.g., communication traces, dependency activations) showing that MARFT leads to more coordinated or efficient agent behaviors?

5. **Scalability and practical feasibility:** The proposed setup requires separate GPUs for each agent and sequential rollouts. How would this approach scale to realistic multi-agent LLM systems with more than three agents or longer tasks? Are there plans for efficient parallelization or off-policy training to address this limitation?

---

> ### Author Response · Authors · 2025-11-23
> **Rebuttal by Authors [Questions/Weaknesses 1&2]**
>
> > **Question/Weakness 1**: Clarification on action dependency modeling: Could the authors provide a concrete example or visualization showing how the dependency function ( D(a_i, a_j) ) is computed or updated during training? For instance, how does one agent’s output affect another’s input in the MARFT framework, and how is this handled across asynchronous timesteps?
>
> **A1**: We appreciate the request for clarification on the practical implementation of the dependency function $D$.
> 1. In our current experiments, the dependency is physically realized through sequential context concatenation. If $D(a^i, a^j) = 1$, the output tokens generated by Agent $i$ are appended to the input context window of Agent $j$. This ensures that Agent $j$'s policy is conditioned on Agent $i$'s actions, naturally handling the information flow.
> 2. While the Flex-MG framework theoretically supports a learnable or dynamic $D$ (where the graph structure itself is optimized), our current work focuses on validating the MARFT optimization algorithm itself. Therefore, in our experiments, $D$ is fixed and pre-defined based on the logical workflow of the specific task (e.g., in the "Duo" setup, the Coder strictly depends on the Reasoner). And drawing from established research in MARL (e.g., Prioritized Multi-Agent Transformer[1]), we acknowledge that while the order does not prevent convergence, it may influence the performance ceiling. In the context of LaMAS, however, the order is often naturally dictated by the logical flow of the task (e.g., a Reasoner must precede a Coder).
> 3. In the Multi-Agent Reinforcement Learning settings, according to the Markov Game, agents take actions in the same timestep, and the asynchronicity only happens when in this timestep, some agents have no dependencies on others, and they can take action asynchronously.
>
> We agree that dynamically optimizing agent orchestration and ordering is a promising direction, e.g., Prioritized Multi-Agent Transformer[1] for LaMAS. Since MARFT is robust to permutation, it provides a stable foundation for future research into learnable or dynamic agent ordering.
>
> > **Question/Weakness 2**: Justification for multi-agent design in math and coding tasks: Why are tasks like MATH and CodeForces appropriate for evaluating multi-agent reinforcement learning? Can the authors demonstrate any scenario where coordination between agents (e.g., Reasoner–Actor or Coder–Reviewer) is essential to solving the task, rather than just increasing model complexity?
>
> **A2**: We appreciate this candid feedback regarding the evaluation of the multi-agent formulation. We offer the following perspective to clarify our experimental design and the current state of the field.
> 1. The primary objective of this work is to validate the theoretical feasibility of the MARFT paradigm. We selected math and coding benchmarks specifically because they provide clear, verifiable rewards, which are essential for rigorously testing a new RL-based optimization method. We also note that relevant baselines like MAPoRL[2] and MARTI[3] similarly rely on mathematical reasoning tasks for validation, establishing this as a standard testbed for current LaMAS research.
> 2. Insightfully, you raise a critical "pain point" in the transition from traditional MARL to LaMAS and this is also stated in the Limitation section. In traditional MARL, environments (e.g., StarCraft) inherently demand cooperation. In contrast, the LaMAS field is still in an early stage where "native" multi-agent benchmarks, which strictly require heterogeneous agents, are scarce.
> 3. For the scenarios where coordination between agents is essential to solving the task, we have added some case studies, which are generated from the CodeForces MARFT experiments in Appendix E. And hope the examples make sense for you.
>
> While current benchmarks can be solved by single agents, our results show that the multi-agent formulation still yields performance gains over single agents. And our extra comparison experiments also preliminarily demonstrate that for more complex tasks, multi-agent formulation, as well as MARFT, can further improve the performance, though the whole community is struggling with the LaMAS environments and benchmarks.

---

> > ### Author Response · Authors · 2025-11-23
> > **Continue: Rebuttal by Authors [Questions/Weaknesses 3&4]**
> >
> > > **Question 3**: Connection and difference with RLHF: The paper repeatedly refers to “reinforcement fine-tuning,” but appears to use standard PPO without human feedback. Could the authors clearly articulate how MARFT differs from RLHF in terms of training signal, objective function, or supervision source?
> >
> > **A3**: We appreciate the opportunity to clarify the terminology and the fundamental distinctions between MARFT and RLHF. While both approaches often utilize PPO as the underlying optimization engine, they diverge significantly in their training signals, objectives, and sources of supervision:
> > 1. As for the training signal, RLHF typically relies on a learned Reward Model (RM) trained on subjective human preferences (e.g., helpfulness, safety). In contrast, MARFT operates in the domain of Reinforcement Learning from Verifiable Rewards (RLVR). The supervision comes from ground-truth, verifiable signals provided by the environment (e.g., passing unit tests in coding or deriving the correct answer in math), rather than a proxy reward model.
> > 2. The objective functions are similar, but the problem is different. In RLHF, the model is optimized to obtain a high reward from the RM, which is trained by human preference data, so it aligns with the human preference score. But in MARFT, it is solving a multi-agent decision-making problem, and is to optimize the agents to better take actions to help the team get a high reward from an environment with verifiable rewards. Though both of these use PPO-like objectives, the core problem statements are totally different.
> > 3. Unlike the independent updates in standard RLHF, MARFT-A employs a sequential trust-region update scheme (similar to MAT/HAPPO in traditional MARL) to ensure monotonic improvement for the collaborative group, addressing the non-stationarity inherent in multi-agent learning.
> >
> > We hope our response effectively addresses your concern. We would be happy to provide clarifications regarding the problem statement or any other aspects of our work during the discussion period.
> >
> >
> > > **Question 4**: Empirical evidence of coordination benefits: Beyond modest accuracy gains, is there any qualitative or behavioral analysis (e.g., communication traces, dependency activations) showing that MARFT leads to more coordinated or efficient agent behaviors?
> >
> > **A4**: We appreciate this suggestion. While our primary objective with MARFT is to establish a rigorous optimization framework based on Markov Games rather than to conduct a sociological study of agent societies, we agree that behavioral evidence is valuable. We address this in three ways:
> > 1. Within the Flex-MG formulation, the agents are strictly interdependent (e.g., the Coder basically cannot succeed if the Reasoner fails). Therefore, the monotonically increasing team reward (as shown in our learning dynamics) serves as the primary, quantitative evidence of improved coordination. The system is mathematically optimizing for the joint success of the workflow.
> > 2. To directly address your request for behavioral insights, we have added Appendix E to the revised manuscript. This section includes extended learning dynamics and concrete case studies. These qualitative examples illustrate how MARFT-tuned agents learn to play their roles well and coordinate.

---

> > > ### Author Response · Authors · 2025-11-23
> > > **Continue: Rebuttal by Authors [Question/Weakness 5 & references]**
> > >
> > > > **Question 5**: Scalability and practical feasibility: The proposed setup requires separate GPUs for each agent and sequential rollouts. How would this approach scale to realistic multi-agent LLM systems with more than three agents or longer tasks? Are there plans for efficient parallelization or off-policy training to address this limitation?
> > >
> > > **A5**: We acknowledge the computational and decentralized challenges inherent in multi-agent fine-tuning. We address the scalability and future directions as follows:
> > > 1. While MARFT is theoretically scalable to any number of agents, practical implementation is currently limited by the lack of established environments and benchmarks specifically for LaMAS that require or support more than three agents. This is a challenge shared by the broader LaMAS (or LLM Multi-Agent) community.
> > > 2. We emphasize that MARFT is a general paradigm based on the Flex-MG formulation. While our current instantiation uses PPO (on-policy), which requires sequential rollouts, the framework is compatible with off-policy algorithms. To address scalability, future work could replace PPO with specific methods like Multi-Agent Soft Actor-Critic (MASAC)[4], where training is centralized but inference can be in a decentralized manner using only local observations, removing the bottleneck of sequential dependency..
> > >
> > > Thus, theoretically MARFT paradigm is compatible with off-policy algorithms, but the implementations of those kinds of methods, like SAC, have not been well explored in the LLM agent domain, which is also a promising research direction in the future.
> > >
> > > ---
> > > [1]Hu, Kun, et al. "PMAT: Optimizing Action Generation Order in Multi-Agent Reinforcement Learning". AAMAS 2025.
> > >
> > > [2]Park, Chanwoo, et al. "MAPoRL: Multi-Agent Post-Co-Training for Collaborative Large Language Models with Reinforcement Learning". ACL 2025.
> > >
> > > [3]Zhang, Kaiyan, et al. "MARTI: A Framework for Multi-Agent LLM Systems Reinforced Training and Inference".
> > >
> > > [4]Zhang, Qingrui, el al. "Lyapunov-Based Reinforcement Learning for Decentralized Multi-Agent Control". DAI 2020.

---

### Official Review · Reviewer_Dgkv · 2025-10-28

**Soundness:** 3
**Presentation:** 2
**Contribution:** 3
**Rating:** 4
**Confidence:** 3

**Summary:**

The paper proposes Multi-Agent Reinforcement Fine Tuning (MARFT), which is a framework for optimizing LLM-based multi-agent systems. The authors develop two algorithms: MARFT-A (action-level) and MARFT-T (token-level). The method is theoretically supported and empirically validated.

**Strengths:**

- The formulation is original, and the idea to apply multi-agent RL for LLM agents is novel, to the best of my knowledge.
- The use of Theorem 1 to justify the sequential nature of the system is sound. It provides a better understanding of the approach.
- The empirical study is showing that the prorposed approach is effective.

**Weaknesses:**

- If I understand the setting correctly, independent RL can also be applied.  Namely that each agent is not aware of the existence of the others. This can be a valid baseline to compare with but it is not shown in the experiments. Actually there is no baseline in the experiments other than the vanilla performance.

- Following the previous point,  it is therefore hard to evaluate the proposed approach without baselines. So at this moment I cannot tell how useful it is to formulate it at a multi-agent level (rather than single agents)

- Figure 3 contains way more information than is presented in the main contents. Concepts including Central Critic Head, Buffer, "inst" are not explained.  So I would suggest to either simplify the figure a lot, or provide more explanation in text.

- Minor: certain notations are missing definition. The advantage function in Theorem 1 is not formally defined.

**Questions:**

- How sensitive is MARFT-A to the order of agent updates? In some cases certain agents can running in parallel rather than sequentially. For example, in figure 1, calendar agent and location agent can run at the same time. Although we can still model it as a sequential problem, does agent ordering affect stability or final performance?
- In section 5.5, do you mean the implementation is without vllm and sglang, and is it because they cannot be incorporated due to some technical issues?

---

> ### Author Response · Authors · 2025-11-23
> **Rebuttal by Authors [Weaknesses part]**
>
> > **Weakness 1**: If I understand the setting correctly, independent RL can also be applied. Namely that each agent is not aware of the existence of the others. This can be a valid baseline to compare with but it is not shown in the experiments. Actually there is no baseline in the experiments other than the vanilla performance.
>
> **A1**: Thank you for this valuable suggestion. We carefully considered MAPoRL[1] but found significant structural differences that made direct comparison challenging, as you pointed out. However, we would like to add a relevant baseline to address this.
>
> 1. As for MAPoRL, it is designed for synchronous multi-agent debate, assuming homogeneous agent behavior. In contrast, our LaMAS setup involves asynchronous, heterogeneous roles (e.g., Reasoner $\rightarrow$ Coder/Actor $\rightarrow$ ...), making the MAPoRL framework structurally incompatible with the sequential nature of our tasks.
> 2. To address your concern and provide a fair comparison experiment, we have implemented Independent PPO (IPPO) as an additional baseline in the Section 5.4 in the revised manuscript. Since IPPO serves as the core algorithm for MAPoRL, this allows us to benchmark the MARFT paradigm against the underlying mechanism of these works. And the comparison experiments also demonstate the superiority of MARFT and preliminarily show that MARFT is more scalable when applied to more complicatedly organized LaMAS for agentic tasks. Please see Section 5.1 for more details.
>
>
> > **Weakness 2**: Following the previous point, it is therefore hard to evaluate the proposed approach without baselines. So at this moment I cannot tell how useful it is to formulate it at a multi-agent level (rather than single agents)
>
> **A2**: We appreciate this candid feedback regarding the evaluation of the multi-agent formulation. We offer the following perspective to clarify our experimental design and the current state of the field.
> 1. The primary objective of this work is to validate the theoretical feasibility of the MARFT paradigm. We selected math and coding benchmarks specifically because they provide clear, verifiable rewards, which are essential for rigorously testing a new RL-based optimization method. We also note that relevant baselines like MAPoRL[1] and MARTI[2] similarly rely on mathematical reasoning tasks for validation, establishing this as a standard testbed for current LaMAS research.
> 2. You raise a critical "pain point" in the transition from traditional MARL to LaMAS and this is also stated in the Limitation section. In traditional MARL, environments (e.g., StarCraft) inherently demand cooperation. In contrast, the LaMAS field is still in an early stage where "native" multi-agent benchmarks, which strictly require heterogeneous agents, are scarce.
>
> While current benchmarks can be solved by single agents, our results show that the multi-agent formulation still yields performance gains over single agents. And our extra comparison experiments also preliminarily demonstrates that for more complex tasks, multi-agent formulation, as well as MARFT, can further improve the performance, though the whole community is struggling with the LaMAS environments and benchmarks.
>
> > **Weakness 3**: Figure 3 contains way more information than is presented in the main contents. Concepts including Central Critic Head, Buffer, "inst" are not explained. So I would suggest to either simplify the figure a lot, or provide more explanation in text.
>
> **A3**: Thank you for your valuable suggestions! We have added more detailed explanations in the caption of the figure in the manuscript.
>
> 1. For the Central Critic Head, it is an extra trainable multilayer perceptron (MLP) with parameters $\phi$ to map the hidden vectors output by the frozen transformer module to specific values.
> 2. For the Buffer, when the LaMAS is interacting with the environment and generating trajectories, the trajectory data will be stored in the ReplayBuffer. Then, during training, the data will be used to compute the values for optimization. Since the data in the Buffer will be discarded once used, the Buffer is an on-policy buffer.
> 3. For the $\text{inst}^i$, it is the agent's specific system prompt/profile. And we provided the profiles configurations in the Appendix D.3.
>
> If there is anything else not being well explained, please let us know.
>
> > **Weakness 4**: Minor: certain notations are missing definition. The advantage function in Theorem 1 is not formally defined.
>
> **A4**: We apologize for the confusion caused by the unclear definitions of some notations. According to your suggestion, we added necessary definitions and detailed explanations for Theorem 1. Your valuable advice is helping make the paper more complete.

---

> > ### Author Response · Authors · 2025-11-23
> > **Continue: Rebuttal by Authors [Questions part]**
> >
> > > **Question 1**: How sensitive is MARFT-A to the order of agent updates? In some cases certain agents can running in parallel rather than sequentially. For example, in figure 1, calendar agent and location agent can run at the same time. Although we can still model it as a sequential problem, does agent ordering affect stability or final performance?
> >
> > **A5**: This is a very insightful question. We explain the sensitivity to agent order from both theoretical and practical perspectives.
> > 1. As stated in Theorem 1 (Multi-Agent Advantage Decomposition Theorem), the decomposition holds for any predefined permutation of agents. Consequently, the optimization objective remains mathematically valid, and the algorithm is guaranteed to converge regardless of the execution order.
> > 2. Drawing from established research in MARL (e.g., Prioritized Multi-Agent Transformer), we acknowledge that while the order does not prevent convergence, it may influence the performance ceiling. In the context of LaMAS, however, the order is often naturally dictated by the logical flow of the task (e.g., a Reasoner must precede a Coder).
> >
> > We agree that dynamically optimizing agent orchestration and ordering is a promising direction. Since MARFT is robust to permutation, it provides a stable foundation for future research into learnable or dynamic agent ordering.
> >
> >
> >
> > > **Question 2**: In section 5.5, do you mean the implementation is without vllm and sglang, and is it because they cannot be incorporated due to some technical issues?
> >
> > **A6**: Yes, in the preliminary experiments, we are running them using a very simple framework without vLLM or SGLang. We have preliminarily implemented MARFT based on well-established codebase, like AReal[3] (AReal is implemented with SGLang), based on which, we ran the supplementary comparison experiments. There is indeed some technical issues. For example, vLLM purely provides response generation service, while SGLang provides more complex and agent-style service, like multi-turn, tool calling, etc. Additionally, how to well deal with the parallel trajectory collection can also be a tricy problem, because in real RL process, the episode length varies. If we frequently collect trajectories in parallel for big batches, this kind of data irregularity leads to significant computational inefficiency. Specifically, when batching trajectories of varying lengths (i.e., ragged batches), standard synchronous execution suffers from severe padding overheads and synchronization barriers, where the GPU sits idle waiting for the longest episode to complete.
> >
> > So from the technical challenges, we also see the urgent need for a better framework, codebase or even inference engine specifically for LaMAS RFT. We believe this could also be a very significant research direction, which potentially will push the RFT for LaMAS to another level.
> >
> > ---
> > [1]Park, Chanwoo, et al. "MAPoRL: Multi-Agent Post-Co-Training for Collaborative Large Language Models with Reinforcement Learning". ACL 2025.
> >
> > [2]Zhang, Kaiyan, et al. "MARTI: A Framework for Multi-Agent LLM Systems Reinforced Training and Inference".
> >
> > [3]Fu, Wei, et al. "AReaL: A Large-Scale Asynchronous Reinforcement Learning System for Language Reasoning".

---

### Official Review · Reviewer_ojuV · 2025-10-31

**Soundness:** 3
**Presentation:** 3
**Contribution:** 3
**Rating:** 6
**Confidence:** 3

**Summary:**

This paper introduces MARFT (Multi-Agent Reinforcement Fine-Tuning), a new paradigm for optimizing LLM-based Multi-Agent Systems (LaMAS) via reinforcement learning. The authors first formalize the Flexible Markov Game (Flex-MG) to account for agent heterogeneity, asynchronous execution, and dynamic organizational dependencies. Building upon this, they propose MARFT-A (action-level) and MARFT-T (token-level) instantiations, extending PPO-like optimization to multi-agent language systems. Experiments on math problem solving and coding tasks demonstrate consistent performance improvements over vanilla LaMAS baselines.

**Strengths:**

1. The paper generalizes reinforcement fine-tuning from single-agent MARL to LaMAS systems, addressing an important theoretical gap.
2. The proposed Flex-MG formulation is reasonable and effectively models dynamic dependencies among agents.
3. The experiments on coding and math problem-solving tasks verify the effectiveness of the proposed method, particularly MARFT-A, which shows stable and consistent improvement.

**Weaknesses:**

1. The method and experiments are limited to single-round tasks, which raises concerns about the framework’s scalability to more complex or multi-turn interactive environments. Could MARFT be extended to handle richer LaMAS settings (e.g., multi-turn reasoning or tool-use workflows)?
2. The paper lacks comparison with relevant RL-based LaMAS works, such as MAPoRL[1].
[1] Park, Chanwoo, et al. "Maporl: Multi-agent post-co-training for collaborative large language models with reinforcement learning." arXiv preprint arXiv:2502.18439 (2025).

**Questions:**

Please see weakness.

---

> ### Author Response · Authors · 2025-11-23
> **Rebuttal by Authors**
>
> > **Weakness 1**: The method and experiments are limited to single-round tasks, which raises concerns about the framework’s scalability to more complex or multi-turn interactive environments. Could MARFT be extended to handle richer LaMAS settings (e.g., multi-turn reasoning or tool-use workflows)?
>
> **A1**: We appreciate the opportunity to clarify the scope of our experiments and the capabilities of the MARFT framework.
>
> 1. We respectfully point out that our mathematical reasoning experiments were indeed conducted in a multi-turn setting (which is, though a little bit limited). To quantify performance in this context, we introduced the Average Step Reward (ASR) metric, calculated by dividing the episodic return by the number of steps taken. And our results demonstrate that MARFT successfully optimizes for multi-turn interactions. As shown in our analysis of learning dynamics, the agents demonstrated a constant improvement in ASR, indicating that the system learned not only to solve problems correctly but also to do so more efficiently by optimizing the number of interaction turns.
> 2. Theoretically, the MARFT paradigm is designed to extend to richer LaMAS settings (such as complex tool use) without modification. However, the practical limitation is computational rather than algorithmic. Scaling to longer, open-ended interactions introduces significant costs due to expanding context windows and the need for larger models. We view addressing these resource bottlenecks as a critical next step for the field, alongside the development of more comprehensive LaMAS environments and benchmarks.
>
> Scaling RFT for LaMAS to larger and more complex interactive environments is definitely promising but challenging future research direction, which requires way more resources and infrastracture support.
>
>
> > **Weakness 2**: The paper lacks comparison with relevant RL-based LaMAS works, such as MAPoRL.
>
> **A2**: We appreciate the reviewer highlighting MAPoRL[1] as a relevant related work. We have carefully analyzed MAPoRL and would like to clarify why a direct framework comparison is structurally infeasible, and how we have addressed the underlying algorithmic comparison:
> 1. MAPoRL is designed specifically for synchronous, homogeneous settings, primarily focusing on collaborative debate where agents share similar roles and act simultaneously. In contrast, MARFT is tailored for asynchronous, heterogeneous LaMAS workflows (e.g., Reasoner $\rightarrow$ Coder/Actor $\rightarrow$ ...), where agents must strictly adhere to distinct profiles and sequential dependencies. And this also means MARFT is more scalable, as MAPoRL mainly focuses on multi-agent debate.
> 2. Because the framework of MAPoRL relies on assumptions (synchronicity and homogeneity) that do not hold in our functional agentic tasks, directly applying it is not possible. However, recognizing that MAPoRL is algorithmically grounded in Independent PPO (IPPO), we have implemented IPPO as a new baseline in Section 5.2 in our revised manuscript. This allows us to provide a fair and direct comparison against the core optimization mechanism used in MAPoRL. And the comparison experiments also demonstrate the superiority of MARFT and preliminarily show that MARFT is more scalable when applied to more complicatedly organized LaMAS for agentic tasks. Please see Section 5.2 for more details.
>
> ---
>
> [1]Park, Chanwoo, et al. "MAPoRL: Multi-Agent Post-Co-Training for Collaborative Large Language Models with Reinforcement Learning". ACL 2025.

---

### Official Review · Reviewer_AzVe · 2025-10-31

**Soundness:** 2
**Presentation:** 3
**Contribution:** 3
**Rating:** 4
**Confidence:** 2

**Summary:**

This paper introduces MARFT (Multi-Agent Reinforcement Fine-Tuning), a novel framework that extends reinforcement fine-tuning from single-agent language models to multi-agent LLM systems.The authors formalize a new theoretical model called Flexible Markov Game (Flex-MG) to represent asynchronous, dependency-driven agent interactions, and prove the Multi-Agent Advantage Decomposition Theorem, which allows global rewards to be decomposed into per-agent advantages.Two concrete instantiations are implemented and evaluated on reasoning (MATH, CMATH, GSM8K) and coding (CodeForces) environments. Results show that MARFT significantly improves multi-agent collaboration performance over standard supervised fine-tuning baselines.

**Strengths:**

- The paper systematically extends RL-based fine-tuning into the multi-agent LLM regime, which has not been thoroughly studied before.
- The Advantage Decomposition Theorem provides a clean bridge between joint and sequential optimization, addressing credit assignment across agents.-
- Well-written and well-organized

**Weaknesses:**

- Limited experimental scope. Experiments are restricted to relatively small-scale environments and specific LLMs (mainly Qwen-based), which raises questions about generalization to larger or open-ended LaMAS systems.
- While MARFT is compared to SFT-based LaMAS baselines, it would be valuable to include multi-agent RL algorithms for a more comprehensive comparison, especially MARTI and MAPoRL, which were mentioned in the introduction.

**Questions:**

- How sensitive is MARFT-A to the order of agents (since the framework assumes sequential dependencies)?

---

> ### Author Response · Authors · 2025-11-23
> **Rebuttal by Authors**
>
> > **Weakness 1**: Limited experimental scope. Experiments are restricted to relatively small-scale environments and specific LLMs (mainly Qwen-based), which raises questions about generalization to larger or open-ended LaMAS systems.
>
> **A1**: Thank you for pointing this out. We acknowledge the concern regarding the scope of experiments. We would like to offer two clarifications regarding our design choices:
> 1. For the concerns about the environments, as discussed in our Limitations (Section 7), there is currently a significant lack of large-scale benchmarks specifically designed for LaMAS. We selected Math and Coding environments as they provide established, reliable metrics to validate the fundamental effectiveness of the MARFT paradigm without introducing the noise of unverified benchmarks.
> 2. Multi-agent co-training is computationally intensive and we selected Qwen-based models because they seem to offer better instruction-following capabilities to maintain distinct agent roles (critical for our "Duo" and "Trio" setups) while remaining within a manageable parameter count (<10B) for academic research resources. While we tested on specific models, MARFT is derived from the Multi-Agent Advantage Decomposition Theorem (Theorem 1). This theoretical foundation ensures that the optimization method itself is model-agnostic and should generalize to larger architectures, provided sufficient compute is available.
>
> We agree that developing broader LaMAS benchmarks is a critical direction for the community, as pointed out in your comment and emphasized in our conclusion.
>
> > **Weakness 2**: While MARFT is compared to SFT-based LaMAS baselines, it would be valuable to include multi-agent RL algorithms for a more comprehensive comparison, especially MARTI and MAPoRL, which were mentioned in the introduction.
>
> **A2**:  Thank you for this valuable suggestion. We carefully considered both MARTI[1] and MAPoRL[2] but found significant structural differences that made direct comparison challenging. However, we would like to add a relevant baseline to address this.
>
> 1. About MARTI, as noted in our related works, MARTI operates fundamentally as a framework for independent agent training (similar to Independent PPO), where actors with their separate critics optimize individual rewards via specific verifiers rather than a shared team reward. This diverges from our focus on co-fine-tuning a cohesive team towards a global objective.
> 2. As for MAPoRL, it is designed for synchronous multi-agent debate, assuming homogeneous agent behavior. In contrast, our LaMAS setup involves asynchronous, heterogeneous roles (e.g., Reasoner $\rightarrow$ Coder/Actor $\rightarrow$ ...), making the MAPoRL framework structurally incompatible with the sequential nature of our tasks.
> 3. To address your concern and provide a fair comparison experiment, we have implemented Independent PPO (IPPO) as an additional baseline in the Section 5.4 in the revised manuscript. Since IPPO serves as the core algorithm for both MARTI and MAPoRL, this allows us to benchmark the MARFT paradigm against the underlying mechanism of these works. And the comparison experiments also demonstate the superiority of MARFT and preliminarily show that MARFT is more scalable when applied to more complicatedly organized LaMAS for agentic tasks. Please see Section 5.2 for more details.
>
> > **Qestion 1**: How sensitive is MARFT-A to the order of agents (since the framework assumes sequential dependencies)?
>
> **A3**: This is an insightful question. We address the sensitivity to agent order from both theoretical and practical perspectives.
> 1. As stated in Theorem 1 (Multi-Agent Advantage Decomposition Theorem), the decomposition holds for any predefined permutation of agents. Consequently, the optimization objective remains mathematically valid, and the algorithm is guaranteed to converge regardless of the execution order.
> 2. Drawing from established research in MARL (e.g., Prioritized Multi-Agent Transformer[3]), we acknowledge that while the order does not prevent convergence, it may influence the performance ceiling. In the context of LaMAS, however, the order is often naturally dictated by the logical flow of the task (e.g., a Reasoner must precede a Coder).
>
> We agree that dynamically optimizing agent orchestration and ordering is a promising direction. Since MARFT is robust to permutation, it provides a stable foundation for future research into learnable or dynamic agent ordering.
>
> ---
> [1]Zhang, Kaiyan, et al. "MARTI: A Framework for Multi-Agent LLM Systems Reinforced Training and Inference".
>
> [2]Park, Chanwoo, et al. "MAPoRL: Multi-Agent Post-Co-Training for Collaborative Large Language Models with Reinforcement Learning". ACL 2025.
>
> [3]Hu, Kun, et al. "PMAT: Optimizing Action Generation Order in Multi-Agent Reinforcement Learning". AAMAS 2025.

---

### Author Response · Authors · 2025-11-23
**Meta response about the rebuttal and the revision of the paper**

Dear Area Chairs and Reviewers,

We thank Reviewers AzVe, ojuV, Dgkv, and xPkj for their constructive and insightful feedback. Your contributions have significantly strengthened our paper, leading to substantial revisions in empirical evaluation, theoretical clarity, and architectural explanation.

1. Enhanced Empirical Rigor and Baselines (Addressing concerns from Reviewers AzVe, ojuV, and Dgkv regarding baseline comparisons):
    - Baseline Implementation: We implemented IPPO (Independent PPO) in Section 5.2 as a critical MARL baseline. This serves as the core comparison against independent training frameworks (like MARTI/MAPoRL), demonstrating that $\mathbf{MARFT}$'s co-fine-tuning mechanism is superior for asynchronous, sequential LaMAS tasks.
    - Structural Justification: We clarified that direct comparison with synchronous debate models (like MAPoRL) was challenging due to the inherent asynchronous and heterogeneous nature of our LaMAS setup. The new results firmly validate $\mathbf{MARFT}$'s design for these complex interactions.
2. Theoretical and Architectural Clarity (We strengthened the theoretical presentation, addressing points from Reviewers Dgkv and xPkj):
    - Formal Definitions: We provided detailed definitions for the value and advantage functions used in the Multi-Agent Advantage Decomposition Theorem (Theorem 1) in Section 3, resolving the ambiguity noted by Reviewer Dgkv.
    - Figure Explanation: We enhanced the explanation of Figure 3, clarifying the roles of the Central Critic Head and the Buffer components, making the overall architecture clearer.
    - Dependency Function ($D(a_i, a_j)$) (Reviewer xPkj): We clarified that the dependency is captured explicitly via the sequential execution and shared context, where the output of one agent becomes the constrained input observation for the subsequent agent, making the mechanism concrete in implementation.

3. Agent Dynamics and Conceptual Positioning (Addressing questions on model generalization and conceptual boundaries from Reviewers AzVe, ojuV, and xPkj):
    - Agent Order Sensitivity (Reviewers AzVe, Dgkv): We confirmed that Theorem 1 guarantees convergence for any predefined permutation, ensuring theoretical robustness. Practically, we noted that in LaMAS, the order is often logically dictated by the task (e.g., Reasoner $\rightarrow$ Coder). Dynamic ordering remains a promising direction for future work.
    - RLHF Distinction (Reviewer xPkj): We clarified that $\mathbf{MARFT}$ is a fundamentally distinct paradigm from RLHF. While both may employ algorithms like PPO, RLHF is focused on aligning a single agent with human preferences via a Reward Model. In contrast, $\mathbf{MARFT}$ is explicitly designed for the co-fine-tuning of multiple, heterogeneous, interdependent agents towards a global, task-completion objective. The core novelty of $\mathbf{MARFT}$ lies in the Flex-MG formalization and the Multi-Agent Advantage Decomposition Theorem in the LaMAS domain, which are dedicated mechanisms for credit assignment in complex multi-agent collaboration, pioneering and extending traditional MARL to the LaMAS domain.
    - Scope and Generalization (Reviewers AzVe, ojuV): We acknowledged the current focus on specific LLMs and limited tasks due to computational limits and the lack of robust large-scale LaMAS benchmarks. We reiterated that $\mathbf{MARFT}$’s theoretical foundation is model-agnostic and should generalize.

We believe these revisions address all major concerns and firmly establish $\mathbf{MARFT}$ as a principled and superior framework for fine-tuning LLM-based Multi-Agent Systems.

---

### Meta-Review · Area_Chair_LT1W · 2026-01-01

**Summary:**

Although the paper proposes a theoretically grounded Multi-Agent Reinforcement Fine-Tuning (MARFT) framework, I recommend rejection because the reviewers consistently found the experimental settings (artificial multi-agent roles for standard math/coding tasks) unconvincing and the empirical gains insufficient to justify the added complexity over single-agent baselines.

**Reviewer Concerns:**

The authors' rebuttal successfully clarified the theoretical definitions (Theorem 1) and added a requested IPPO baseline, but the fundamental concern raised by Reviewers xPkj and AzVe—that the proposed multi-agent paradigm is forced onto tasks that do not inherently require coordination—remains a critical, outstanding issue.

**Reviewer Scores:**

Reviewer AzVe may marginally raised their score (from 4) given the new baselines, but Reviewer xPkj may likely maintain their strong rejection (3) as the rebuttal failed to address their core conceptual objection regarding the validity of the research premise.

---

### Decision · Program_Chairs · 2026-01-26

Reject